Progressive overload without progressing load? The effects of load or repetition progression on muscular adaptations

Plotkin Daniel 1
Coleman Max 1
Van Every Derrick 1
Maldonado Jaime 1
Oberlin Douglas 1
Israetel Michael 2
Feather Jared 2
Alto Andrew 1
http://orcid.org/0000-0003-3166-0688 Vigotsky Andrew D. 3
http://orcid.org/0000-0003-4979-5783 Schoenfeld Brad J. 1 bradschoenfeldphd@gmail.com
1 City University of New York, Herbert H. Lehman College , Bronx , United States
2 Renaissance Periodization , Charlotte, NC , United States
3 Northwestern University , Evanston , United States
Clemente Filipe
Electronic publication date: 2022 Sep 30
Publication date: 2022
Volume: 10
Electronic Location ID: e14142
Received 2022 Jun 30; Accepted 2022 Sep 7
Copyright: © 2022 Plotkin et al.
Copyright year: 2022
Copyright holder: Plotkin et al.
License: This is an open access article distributed under the terms of the Creative Commons Attribution License, which permits unrestricted use, distribution, reproduction and adaptation in any medium and for any purpose provided that it is properly attributed. For attribution, the original author(s), title, publication source (PeerJ) and either DOI or URL of the article must be cited.
License URL: https://creativecommons.org/licenses/by/4.0/

Keywords: Progressive overload, Specificity, Muscular adaptations, Muscle hypertrophy, Muscle strength

Funding: PSC CUNY grant from the State of New York 64346-00 52 This study was supported by a PSC CUNY grant from the State of New York (Award # 64346-00 52). The funders had no role in study design, data collection and analysis, decision to publish, or preparation of the manuscript.

==============================
Background

Progressive overload is a principle of resistance training exercise program design that typically relies on increasing load to increase neuromuscular demand to facilitate further adaptations. However, little attention has been given to another way of increasing demand—increasing the number of repetitions.

Objective

This study aimed to compare the effects of two resistance training programs: (1) increasing load while keeping repetition range constant vs (2) increasing repetitions while keeping load constant. We aimed to compare the effects of these programs on lower body muscle hypertrophy, muscle strength, and muscle endurance in resistance-trained individuals over an 8-week study period.

Methods

Forty-three participants with at least 1 year of consistent lower body resistance training experience were randomly assigned to one of two experimental, parallel groups: A group that aimed to increase load while keeping repetitions constant (LOAD: n = 22; 13 men, nine women) or a group that aimed to increase repetitions while keeping load constant (REPS: n = 21; 14 men, seven women). Subjects performed four sets of four lower body exercises (back squat, leg extension, straight-leg calf raise, and seated calf raise) twice per week. We assessed one repetition maximum (1RM) in the Smith machine squat, muscular endurance in the leg extension, countermovement jump height, and muscle thickness along the quadriceps and calf muscles. Between-group effects were estimated using analyses of covariance, adjusted for pre-intervention scores and sex.

Results

Rectus femoris growth modestly favored REPS (adjusted effect estimate (CI90%), sum of sites: 2.8 mm [−0.5, 5.8]). Alternatively, dynamic strength increases slightly favored LOAD (2.0 kg [−2.4, 7.8]), with differences of questionable practical significance. No other notable between-group differences were found across outcomes (muscle thicknesses, <1 mm; endurance, <1%; countermovement jump, 0.1 cm; body fat, <1%; leg segmental lean mass, 0.1 kg), with narrow CIs for most outcomes.

Conclusion

Both progressions of repetitions and load appear to be viable strategies for enhancing muscular adaptations over an 8-week training cycle, which provides trainers and trainees with another promising approach to programming resistance training.

Introduction

Resistance training (RT) is a powerful tool to aid in developing muscle size, strength, endurance, power, and many other positive physiological outcomes (Kraemer, Ratamess & French, 2002). To facilitate the continuation of positive adaptations, a given training regimen must contain some form of progression for a given stimulus (Kraemer, Ratamess & French, 2002). Maintaining a sufficient stimulus to match adaptive capacity is termed progressive overload. Although progressive overload can be applied across an array of progression schemes and periodization models, current progression models generally involve some form of load manipulation (Suchomel et al., 2021).

Load, defined as the magnitude of mass lifted, modifications through a training cycle have historically been accompanied by a change in another variable such as sets, repetitions, velocity, and perceived fatigue (Balsalobre-Fernández & Torres-Ronda, 2021; Lorenz & Morrison, 2015; Helms et al., 2016). While the term progressive overload refers to “the gradual increase of stress placed on the body during resistance training” (Kraemer, Ratamess & French, 2002), the common assumption is that there will be some form of load progression as part of a training regimen. Indeed, traditional progression models attempt to progress load mainly by manipulating the relationship between set volume and intensity of load, while typically rendering prescriptions as a percentage of one-repetition maximum (1RM) (Lorenz & Morrison, 2015). From periodization models to autoregulation and velocity-based training, load is the principal variable that is manipulated (Matveyev, 1977).

While there is little question that manipulating load is a viable strategy for accomplishing many or most training objectives, current evidence indicates that similar hypertrophic outcomes can occur across a wide spectrum of loading ranges (i.e., between five and 30 or more repetitions), provided that sets are equated and are carried out with a high degree of effort (Schoenfeld et al., 2017). Moreover, although there appears to be some credence to the presence of a strength-endurance continuum, with greater strength increases observed with heavier loads and greater muscular endurance improvements with lighter loads, the extent of differences between conditions remains somewhat equivocal (Schoenfeld et al., 2021). Given this knowledge, the question arises as to whether load progressions are necessary to maximize hypertrophy, particularly in the context of relatively short-term training cycles within a training career. Current evidence has compared training outcomes between groups that maintain a certain rep range (i.e., high, moderate, or low). Thus, it is unclear whether load or repetition progressions through a training cycle would elicit differential hypertrophic outcomes. This study aimed to compare the effects of load increases while keeping repetition range constant vs increasing repetitions while keeping load constant on measures of lower body muscle hypertrophy, strength, jump performance, and local endurance in resistance-trained individuals over an 8-week study period. We hypothesized that effort (proximity to failure) and volume (number of working sets) are of principal importance for hypertrophic outcomes, implying that hypertrophy would be similar between load and repetition progression models. Due to the hypothesized specificity of strength adaptations, we predicted that load progressions would produce superior maximum strength and that repetition progressions would produce better muscular endurance due to the available literature on the repetition continuum and the principle of specific adaptations to imposed demands (Schoenfeld et al., 2015; Anderson & Kearney, 1982).

Materials and Methods

Participants

We recruited a convenience sample of 43 resistance-trained volunteers (27 men, 16 women) from a university population (height = 169.5 ± 10.5 cm; body mass = 77.2 ± 16.7 kg; body fat = 23.6% ± 9.5%; age = 23.1 ± 5.3 years; training experience = 3.8 ± 4.0 years). As previously described (Schoenfeld et al., 2019), this sample size was justified by an a priori precision analysis for the minimum detectable change at the 68% level (MDC68%; i.e., 1SD, which is conservative in that it requires a larger sample to produce a narrow interval) for mid-thigh thickness (i.e., SEM×2=2.93mm), such that the compatibility interval (CI) of the between-group effect would be approximately ±MDC68%. Based on data from previous research (Schoenfeld et al., 2019), along with their sampling distributions, Monte Carlo simulation was used to generate 90% CI widths for 5,000 random samples of each sample size. To ensure a conservative estimate, as literature values may not be extrapolatable, the sum of each simulated sample size’s 90% CI’s mean and standard deviation was used, and the smallest sample that exceeded MDC68% was chosen; that is, 18 participants per group (1:1 allocation ratio). Additional participants were recruited to account for the possibility of dropout.

To qualify for inclusion in the study, participants were required to be: (a) between the ages of 18–35 years; (b) free from existing cardiorespiratory or musculoskeletal disorders; (c) self-reported as free from consumption of anabolic steroids or any other legal or illegal agents known to increase muscle size currently and for the previous year; and, (d) considered as resistance-trained, defined as consistently lifting weights at least three times per week (on most weeks) for at least 1 year and regularly working the lower body muscles at least once per week. Participants were asked to refrain from the use of alleged muscle-building supplements throughout the course of the study period.

Participants were randomly assigned to one of two experimental, parallel groups: A group that aimed to increase load while keeping repetitions constant (LOAD: n = 22; 13 men, 9 women) or a group that aimed to increase repetitions while keeping load constant (REPS: n = 21; 14 men, seven women). Randomization into groups was carried out using block randomization, with two or four participants per block (randomized for each block), in R software (R Development Core Team, 2019). Approval for the study was obtained from the Lehman College Institutional Review Board (#2021–2132). Written informed consent was obtained from all participants prior to beginning the study. The methods for this study were preregistered prior to recruitment (https://osf.io/yvhcs).

Resistance training procedures

The RT protocol targeted the lower body musculature and consisted of four sets of the free-weight back squat, leg extension, straight-leg calf raise, and seated calf raise. Participants were prescribed the same upper body RT program with a traditional progression model to follow on alternate training days (without supervision from the researchers) and were instructed to refrain from performing any additional lower body RT for the duration of the study.

Prior to training, participants underwent 10RM testing to determine individual initial training loads for each exercise. The RM testing was consistent with recognized guidelines as established by the National Strength and Conditioning Association (Baechle & Earle, 2008). Training for both routines consisted of two weekly sessions performed on non-consecutive days for 8 weeks. The initial training routines (Session 1) for both groups attempted to maintain an 8–12 repetition maximum (RM) per set per exercise. In subsequent sessions, the LOAD group aimed to increase load while maintaining this target repetition range, whereas the REP group aimed to increase the number of repetitions performed per set while maintaining the initial load. As previously described (Schoenfeld et al., 2016), to help standardize the effort of the training protocols, we verbally encouraged participants to perform all sets to the point of momentary concentric muscular failure, herein defined as the inability to perform another concentric repetition while maintaining proper form. Participants were instructed to perform repetitions in a controlled fashion, with a concentric action of approximately 1 s and an eccentric action of approximately 2 s. Participants were afforded 2 min rest between sets. All routines were directly supervised by the research team to monitor proper performance of the respective routines and ensure participant safety.

Dietary adherence

Data were collected similar to as previously described (Schoenfeld et al., 2015). Specifically, to avoid potential dietary confounding of results, participants were advised to maintain their customary nutritional regimens. Dietary adherence was assessed by self-reported 5-day food records (including at least 1 weekend day) using MyFitnessPal.com (http://www.myfitnesspal.com), which has good relative validity for tracking energy and macronutrient intake (Teixeira et al., 2018). Nutritional data were collected twice during the study: 1 week before the first training session (i.e., baseline) and during the final week of the training protocol. Participants were instructed on how to properly record all food items and their respective portion sizes consumed for the designated period of interest. Each item of food was individually entered into the program, and the program provided relevant information as to total energy consumption, as well as the amount of energy derived from proteins, fats, and carbohydrates for each time-period analyzed.

Measurements

The following measurements were conducted pre- and post-study in a separate resting session. Participants reported to the lab having refrained from any exercise other than activities of daily living for at least 48 h prior to baseline testing and at least 48 h prior to testing at the conclusion of the study. Anthropometric and muscle thickness assessments were performed first in the session, followed by measures of muscle strength. Each strength assessment was separated by a half-hour recovery interval to ensure restoration of resources. Subjects were allowed to consume food ad libitum after anthropometric testing.

Anthropometry

Data were collected similar to as previously described (Schoenfeld et al., 2020). Specifically, participants were told to refrain from eating for 8 h prior to testing, eliminate alcohol consumption for 24 h, abstain from strenuous exercise for 24 h, keep fluid consumption to a minimum on the morning of the test and void their bladder immediately before the test. Participants’ height was measured using a stadiometer and body mass was assessed using a calibrated scale. Estimates of percent body fat and leg segmental lean mass (LSLM) were obtained by bioelectrical impedance analysis (InBody 770; InBody, Cerritos, CA, USA).

Muscle thickness

Data were collected similar to as previously described (Schoenfeld et al., 2020; Schoenfeld et al., 2020). Specifically, ultrasound imaging was used to obtain measurements of MT in longitudinal and transverse modes. A trained ultrasound technician performed all testing using a B-mode ultrasound imaging unit (Model E1; SonoScape Co., Ltd, Shenzhen, China). The technician applied a water-soluble transmission gel (Aquasonic 100 Ultrasound Transmission gel; Parker Laboratories Inc., Fairfield, NJ, USA) to each measurement site, and a 4–12 MHz linear array ultrasound probe was placed on the tissue interface without depressing the skin. When the quality of the image was deemed to be satisfactory, the technician saved the image to a hard drive and obtained MT dimensions by measuring the distance from the subcutaneous adipose tissue-muscle interface to either the aponeurosis or the muscle-bone interface. Values for each measure were obtained by using the machine’s calculation package.

Measurements for each respective site were taken with a tape measure on the right side of the body at the mid-quadriceps femoris (a composite of the rectus femoris (RF) and vastus intermedius), lateral quadriceps femoris (a composite of the vastus lateralis (VL) and vastus intermedius), medial gastrocnemius (MG), lateral gastrocnemius (LG), and soleus muscles. Each site was marked with a felt-tip pen to ensure consistency of measures. For the quadriceps, measurements were obtained at 30%, 50%, and 70% between the lateral epicondyle of the femur and greater trochanter. For the calf muscles, measurements were taken on the posterior surface of both legs at 25% of the lower leg length (the distance from the articular cleft between the femur and tibia condyles to the lateral malleolus).

To ensure that swelling in the muscles from training did not obscure MT results, images were obtained at least 48 h after the training sessions both in the pre- and post-study assessment. This is consistent with research showing that acute increases in MT return to baseline within 48 h following a RT session (Ogasawara et al., 2012) and that muscle damage is minimal after repeated exposure to the same exercise stimulus over time (Biazon et al., 2019; Damas et al., 2016). To further ensure accuracy of measurements, three images were obtained for each site and then averaged to obtain a final value. The test-retest intraclass correlation coefficient (ICC) from our lab for muscle thickness measurements are excellent (>0.94) with coefficients of variation (CV) of ≤3.3%.

Countermovement jump

Data were collected similar to as previously described (Schoenfeld et al., 2020). Specifically, the countermovement jump was used as a proxy measure of explosive lower body performance. The participant was instructed on the proper performance of the counter-movement jump. Performance was carried out as follows: The participant assumed a shoulder-width stance with the body upright and hands on hips. When ready to perform the movement, the participant descended into a semi-squat position and then forcefully reversed direction, jumping as high as possible before landing with both feet on the ground.

Assessment of jump performance was carried out using a contact mat (Just Jump, Probotics, Huntsville, AL, USA), which was attached to a hand-held computer that recorded airtime and thereby ascertained the jump height. Participants stood on the mat and performed three maximal-effort countermovement jumps with a 1-min rest period between each trial. The highest jump was recorded as the final value.

Dynamic muscle strength

Data were collected similar to as previously described (Schoenfeld et al., 2015). Specifically, dynamic lower body strength was assessed by 1RM testing in the back squat (1RMSQUAT) exercise performed on a Smith machine (Icarian Fitness Equipment, Sun Valley, CA, USA). Participants reported to the lab having refrained from any exercise other than activities of daily living for at least 48 h prior to baseline testing and at least 48 h prior to testing at the conclusion of the study. 1RM testing was consistent with recognized guidelines as established by the National Strength and Conditioning Association (Baechle & Earle, 2008). In brief, participants performed a general warm-up prior to testing consisting of light cardiovascular exercise lasting approximately 5–10 min. Next, a specific warm-up set of the squat of five repetitions was performed at ~50% 1RM followed by one to two sets of 2–3 repetitions at a load corresponding to ~60–80% 1RM. Participants then performed sets of one repetition of increasing weight for 1RM determination. Three minutes rest was afforded between each successive attempt. Participants were required to reach parallel in the 1RMSQUAT for the attempt to be considered successful; a cord was attached across the squat rack at the point where each participant achieved a parallel squat to guide performance. Confirmation of squat depth was obtained by a research assistant positioned laterally to the participant to ensure accuracy. 1RM determinations were made within five attempts. The ICC from our lab for the Smith machine squat is 0.953 with a CV of 2.8%.

Isometric muscle strength

We intended to carry out isometric strength testing of the knee extensors, as noted in pre-registration. However, due to calibration issues with the dynamometer, results were invalid and thus not reported herein.

Muscle strength-endurance

Lower-body muscular strength-endurance was assessed by performing the leg extension exercise on a plate-loaded machine (Life Fitness, Westport, CT, USA) using 60% of the participant’s initial body mass. Participants sat with their back flat against the backrest and grasped the handles of the unit for support. The backrest was adjusted so that the anatomical axis of the participant’s knee joint aligned with the axis of the unit. Participants placed their shins against the pad attached to the machine’s lever arm, with knees bent at a 90° angle. Participants performed as many repetitions as possible using a full range of motion (90° of leg flexion to full extension) while maintaining a constant tempo of 1-0-1 as monitored by a metronome. The test was terminated when the participant could not perform a complete repetition with proper form. Muscular endurance testing was carried out after assessment of muscular strength to minimize the effects of metabolic stress potentially interfering with performance of the latter.

Blinding

To minimize the potential for bias, we incorporated two levels of blinding into the design and analysis of this study. First, the researcher who obtained the ultrasound measurements was blinded to group allocation. Second, the statistician performed blinded analyses; only after the analyses were completed did the research assistant unveil the correct dataset. We were not able to blind the strength-related tests, and thus cannot completely rule out the potential for bias in these measures.

Statistical analyses

Data were analyzed in R (version 4.2.0) (R Development Core Team, 2019). Neither baseline nor within-group inferential statistics were calculated, as baseline significance testing is inconsequential (Senn, 1994) and within-group outcomes are not the subject of our research question (Bland & Altman, 2011), although we descriptively present within-group changes to help contextualize our findings. The effect of group (LOAD vs REP) on each outcome variable was estimated using linear regression with pre-intervention score included as a nuisance parameter (Vickers & Altman, 2001). In addition, we included sex as a covariate since we stratified by sex. All outcomes were modeled using ordinary least squares, except for muscle endurance, which was modeled using Poisson regression with a log link function since the data are counts. Importantly, the log link function exponentiates the linear predictors such that the estimated effects are multiplicative (e.g., group A performed 1.5-times more repetitions than group B) rather than the additive (e.g., group A performed 10 more repetitions than group B). As such, the results estimated using the Poisson model are presented multiplicatively. Model residuals were qualitatively examined for structure and heteroscedasticity. We computed 90% CIs of the adjusted effects using the bias-corrected and accelerated bootstrap with 5,000 replicates. Rather than relying on traditional null hypothesis significance testing, which has been criticized for its use in the biomedical and social sciences (Amrhein, Greenland & McShane, 2019; McShane et al., 2017), we drew inferences via an estimation approach (Gardner & Altman, 1986). That is, we did not wish to binarize the presence of an effect or no effect; rather, we sought to draw inferences about the magnitude and uncertainty of the effects, whether they were close to zero or otherwise.

Secondary analyses were performed on nutrition data, which were analyzed similarly to the MT and strength data; that is, using multiple regression with group dummy-coded and pre-intervention nutrition scores and sex as covariates of no interest. The results of these secondary analyses are presented using mean adjusted effects and their standard errors.

Finally, we performed leave-one-out sensitivity analyses to assess the potential undue influence of any single participant. To do so, we removed each participant, one at a time, and re-estimated the intervention effect and its bootstrapped CIs without the removed participant. This was repeated for each participant in the sample. Participants with undue influence may bias the point estimate (e.g., if they inflate the effect, the point estimate will decrease when they are removed) and increase the variance (i.e., the effect estimate becomes more precise when they are removed).

Results

Of the initial 43 subjects, 38 completed the study (LOAD: n = 21; REPS: n = 17). Reasons for dropouts were: Personal reasons (n = 2), lack of compliance (n = 2), and training-related injury (n = 1). All participants that completed the study participated in >85% of the total sessions (LOAD: 94.9%; REPS: 95.2%). Figure 1 displays a CONSORT diagram of the data collection process. Table 1 presents the pre/post-study descriptive statistics and adjusted intervention effects.

Figure 1 CONSORT diagram of the data collection process.

Table 1 Strength, performance, body composition, and hypertrophy outcomes.

Measure	LOAD	REPS	Adjusted effect (CI90%)	
Pre	Post	Δ	Pre	Post	Δ	
Squat 1RM (kg)	76.9 ± 29.9	98.7 ± 30.8	21.8 ± 21.2	86.8 ± 27.1	106.2 ± 29.6	19.3 ± 7.7	−2.0 [−7.8 to 2.4]	
CMJ (cm)	41.0 ± 10.0	40.9 ± 9.1	−0.1 ± 2.7	42.9 ± 10.4	42.7 ± 10.2	−0.1 ± 3.4	0.1 [−1.5 to1.7]	
Endurance (repetitions)	15.0 ± 4.4	21.6 ± 4.4	6.6 ± 3.0	17.4 ± 4.7	24.1 ± 7.3	6.8 ± 5.5	1.02 [0.93–1.14]*	
LSLM (kg)	16.5 ± 3.3	16.8 ± 3.3	0.3 ± 0.4	17.0 ± 4.0	17.3 ± 3.9	0.3 ± 0.3	0.1 [−0.1 to 0.3]	
LG (mm)	15.5 ± 3.3	16.9 ± 3.2	1.4 ± 1.2	17.3 ± 2.8	18.3 ± 2.8	1.0 ± 0.8	−0.2 [−0.8 to 0.3]	
MG (mm)	18.1 ± 2.6	19.2 ± 2.5	1.1 ± 1.3	18.5 ± 3.0	20.1 ± 3.9	1.5 ± 2.7	0.5 [−0.4 to 2.2]	
SOL (mm)	15.5 ± 2.8	16.5 ± 3.3	1.1 ± 1.1	17.1 ± 4.2	18.2 ± 4.3	1.1 ± 1.3	0 [−0.6 to 0.7]	
RF 30% (mm)	49.8 ± 9.1	53.3 ± 8.4	3.5 ± 3.7	51.0 ± 9.1	55.2 ± 9.4	4.2 ± 2.0	0.8 [−0.6 to 2.3]	
RF 50% (mm)	39.8 ± 8.0	43.3 ± 7.0	3.4 ± 2.6	42.3 ± 7.9	46.7 ± 8.3	4.4 ± 2.2	1.3 [0–2.4]	
RF 70% (mm)	28.4 ± 6.6	31.6 ± 6.3	3.2 ± 2.5	31.7 ± 7.2	35.4 ± 7.7	3.7 ± 2.1	0.7 [−0.4 to 1.9]	
RF sum (mm)	118.1 ± 22.7	128.2 ± 21.0	10.1 ± 7.9	125.0 ± 23.6	137.3 ± 24.6	12.3 ± 5.1	2.8 [−0.5 to 5.8]	
VL 30% (mm)	44.7 ± 8.1	48.0 ± 8.6	3.3 ± 3.1	48.8 ± 11.6	51.8 ± 10.8	3.0 ± 3.1	0 [−1.5 to 1.5]	
VL 50% (mm)	38.8 ± 7.9	42.7 ± 7.4	3.9 ± 2.7	43.0 ± 10.6	45.9 ± 10.4	2.9 ± 2.3	−0.6 [−1.8 to 0.8]	
VL 70% (mm)	27.6 ± 6.3	31.3 ± 6.5	3.7 ± 2.2	32.4 ± 8.6	36.5 ± 9.2	4.1 ± 2.6	0.4 [−1.0 to 1.6]	
VL sum (mm)	111.0 ± 21.2	121.9 ± 21.2	10.9 ± 7.1	123.4 ± 30.1	133.7 ± 30.0	10.2 ± 6.1	−0.3 [−3.6 to 3.6]	
Body fat (%)	25.0 ± 8.2	24.5 ± 7.9	−0.5 ± 2.7	24.1 ± 10.5	23.3 ± 10.9	−0.8 ± 2.3	−0.4 [−1.7 to1.1]	
Notes:

Adjusted effects are REPS relative to LOAD. Higher/positive values favor REPS.

* Exponentiated effect calculated using a Poisson model; on average, participants in REPS performed 1.02-times more repetitions than LOAD. Abbreviations: RM, repetition maximum; CMJ, countermovement jump; LSLM, leg segmental lean mass; LG, lateral gastrocnemius; MG, medial gastrocnemius; SOL, soleus; RF, rectus femoris; VL, vastus lateralis.

Hypertrophy

The effect of REPS relative to LOAD on MT was negligible across all muscles except the RF, and with tight CIs. When summing the sites of the RF, REPS had an adjusted effect of 2.8 mm, and the data were compatible with values ranging from −0.5 to 5.8 mm (Fig. 2).

Figure 2 Baseline- and sex-adjusted muscle thickness change scores.

We adjusted individuals’ changes in muscle thickness by baseline muscle thickness and sex to better depict the group effects estimated by our statistical models. Increases in muscle thickness can be seen across muscles and groups, with minimal differences between groups, except for the RF, in which the REPS group had modestly greater increases in muscle thickness.

Strength

1RMSQUAT’s point estimate slightly favored LOAD as compared to REPS, with an adjusted effect of 2.0 kg. However, the data were compatible with a wide spread of effects, ranging from 7.8 kg in favor of LOAD to 2.4 kg in favor of REPS (Fig. 3A).

Figure 3 Baseline- and sex-adjusted performance measures change scores.

We adjusted individuals’ changes in performance metrics by baseline scores and sex to better depict the group effects estimated by our statistical models. Improvements in both Smith machine squat 1RM and leg extension repetition counts were apparent but similar between groups. In contrast, changes in countermovement jump (CMJ) performance were equivocal and similar between groups.

Muscle endurance

REPS could perform an estimated 2% more repetitions in the leg extension exercise following the intervention as compared to LOAD. The data were compatible with 7% more repetitions for LOAD to 14% more repetitions for REPS (see Fig. 3B).

Countermovement jump

CMJ showed negligible changes in both LOAD and REPS. The data were compatible with a relatively small range of effects, ranging from 1.5 cm favoring LOAD to 1.7 cm favoring REPS (see Fig. 3C).

Body composition

Body fat showed small changes across the study period, with minimal between-group effects. LSLM estimates largely corroborated the MT measures, with a small point estimate (0.1 kg advantage to REPS) and inconsequential CI (0.1 kg in favor of LOAD to 0.3 kg in favor of REPS) (Fig. 4).

Figure 4 Baseline- and sex-adjusted body composition change scores.

We adjusted individuals’ changes in body composition metrics by baseline scores and sex to better depict the group effects estimated by our statistical models. Changes in body composition were modest, albeit with large variances, and similar between groups.

Dietary changes

Dietary changes were negligible across both LOAD and REPS, with minimal between-group effects (Table 2).

Table 2 Dietary changes across both experimental groups.

	LOAD	REPS	Adjusted effect ± SE	
Pre	Post	Pre	Post	
Fat (g)	68 ± 23	68 ± 20	68 ± 23	69 ± 24	1 ± 6	
Carbohydrates (g)	208 ± 63	207 ± 67	201 ± 58	210 ± 59	9 ± 17	
Protein (g)	99 ± 34	92 ± 35	83 ± 25	91 ± 34	9 ± 10	
Calories	1,840 ± 509	1,805 ± 470	1,736 ± 409	1,835 ± 522	93 ± 143	

Leave-one-out sensitivity analyses

We performed leave-one-out sensitivity analyses for all outcomes to assess whether any single participant strongly influenced the estimated effects. While some individuals were slightly influential in some analyses (e.g., MG muscle thickness), none were sufficiently influential to shift our conclusions (Fig. S1).

Discussion

This is the first study designed to directly compare the effects of progressing repetitions vs load on muscular adaptations. Notably, across almost all outcomes, REPS was generally similar to LOAD, suggesting it may be a viable option that provides trainers and trainees additional option for program design (Halperin et al., 2018). In the ensuing paragraphs, we discuss these results in the context of available evidence and speculate on their potential implications for exercise prescription.

Hypertrophy

Both groups gained appreciable muscle mass over the study period, with pooled mean increases ranging from 6.7% to 12.9% across measurement sites; similar increases were observed between conditions for a majority of MT measurements including the soleus, gastrocnemius, and all 3 VL sites (Table 1; Fig. 2). Overall, these results suggest that, from a hypertrophy standpoint, progressions can be made with load, repetitions, or conceivably a combination of the two over the course of an 8-week training block. The results are generally consistent with the body of literature, which shows similar hypertrophy across a wide spectrum of loading ranges (Schoenfeld et al., 2017).

The similar hypertrophic outcomes observed in our study are in contrast to previous work by Nóbrega et al. (2022), who performed a retrospective analysis using groups from two different studies (Barcelos et al., 2018; Nobrega et al., 2018). Contrary to our findings, their results showed that adjusting load elicited substantially greater increases in muscle cross-sectional area of the VL compared to the group that adjusted repetitions (16.0 ± 4.0% vs 7.9 ± 4.0%, respectively; ES = 2.03 [95% CI: 1.04–3.02]). Several differences between the studies may account for the discordant findings, with perhaps the most important being that Nóbrega et al. (2022) did not employ randomization since it was a retrospective analysis, hindering the ability to draw causal inferences.

Intriguingly, REPS showed a modest superiority for increases in summed MT of the RF (point estimate = 2.8 mm) with CIs ranging from negligible negative effects (−0.5 mm) to relatively large positive effects (5.8 mm); the effects were fairly consistent across proximal, mid and distal sites and were not sensitive to leaving any subject out (Fig. S1). Although the reasons for this finding are not entirely clear, it is possible that higher repetition squat training potentiated greater recruitment of the RF due to heightened accumulated fatigue in the vastii musculature, which henceforth would require greater contribution from the RF toward the end of a set. In contrast, it would likely not be as beneficial for the RF to contribute when squat loads are greater since it would counteract the hip extensors. This hypothesis is purely speculative as we currently lack evidentiary insights into the details of recruitment patterns and fatigue dynamics between the specific contexts. Alternatively, it is possible that the observation was simply due to random chance, especially since the other muscles seemed to have similar growth between conditions. Given the relatively modest magnitude of difference between conditions and that only the RF appeared to benefit from REPS relative to LOAD, this should be considered a preliminary finding that requires replication.

Strength

Increases in 1RMSQUAT were ~20 kg on average across groups, but slightly favored LOAD, with a point estimate of 2 kg, or about a 10% greater increase in LOAD compared to REPS (Table 1; Fig. 3A). However, the CI encapsulated effects ranging from relatively modest negative effects to appreciable positive effects for LOAD (−2.4 and 7.8 kg, respectively), calling into question the meaningfulness of differences. The overall lack of consistent, appreciable differences between conditions is somewhat surprising given that the literature generally indicates a dose-response relationship between the magnitude of load and gains in dynamic muscular strength (Schoenfeld et al., 2017). Although speculative, it is possible that the relatively null findings between conditions can be explained by the fact that 1RM testing was conducted on a Smith machine while training was performed using the free-weight back squat. Consistent with the principle of specificity, there may be less overall carryover between a free-weight squat and a Smith machine squat, particularly given that both groups trained relatively far from their 1RM in this exercise. Hence, neither group conceivably would have developed the specific coordination and skill required to optimize 1RM squat performance on the Smith machine. To avoid inferential ambiguity and provide clarity to the matter, future investigations may benefit from incorporating multiple measures of strength (Buckner et al., 2017). From these data alone, it seems REPS may provide lifters with another option to increase their maximal strength.

Muscular endurance

Leg extension endurance increased by ~7 repetitions across both groups and we observed negligible difference between groups, with a CI containing values of no practical significance (Table 1; Fig. 3B). Previous research is mixed as to the effect of the training load on local muscular endurance with some studies showing a benefit to the use of lighter loads and others showing negligible differences across a wide range of loading conditions (Schoenfeld et al., 2021). Notably, studies that base testing on a fixed submaximal load, as was the case in our study, tend to show similar increases in muscular endurance between heavy and lighter loads (Jessee et al., 2018; Buckner et al., 2019), supporting the notion that REPS and LOAD are both viable options to increase muscular endurance.

Countermovement jump

CMJ performance neither improved nor differed between groups (Table 1; Fig. 3C). In athletic populations, the general observation is that as maximal strength increases relative to body mass, indices of explosive performance improve correspondingly (Nuzzo et al., 2008). However, while our population was trained, they were not necessarily athletic. Thus, the combination of a lack of appreciable differences in strength, the lack of specific jump training, and the given population may explain the lack of changes in either group.

It also should be noted that the emphasis of repetitions in both groups was to control the weight, particularly on the eccentric action, but also during the concentric action as well (cf., maximum concentric velocity). Thus, benefits related to highly dynamic strength, such as the stretch-shortening cycle, may not have been as pronounced. Qualitatively, it was also visibly apparent that many participants lacked the specific coordination for efficient performance of the CMJ, perhaps limiting their ability to exploit the effects of the interventions.

Limitations

Our study had several limitations that should be considered when attempting to draw inferences from the data. First, we tried to account for dietary practices via 5-day food diaries at the beginning and end of the study under the guidance of trained nutrition professionals. While food diaries are a well-accepted method for estimating nutritional consumption, evidence indicates widespread discrepancies between what is reported and what is actually consumed (Mertz et al., 1991). It therefore remains possible that despite our attempts to control nutritional intake, between-group differences in energy- and/or macronutrient-related factors may have confounded results. Although possible, body fat estimates via multifrequency BIA indicated similar changes between REPS and LOAD and results were within the standard error of measurement of the modality (Schoenfeld et al., 2018), suggesting a relative group-level maintenance of body fat over the study period; this indicates total energy intake was likely similar between conditions. Second, our sample comprised young resistance-trained men and women; thus, results cannot necessarily be generalized to other populations including adolescents, older individuals, and untrained populations. Third, training and testing were specific to the calves and quadriceps, thus inferences cannot be drawn for other lower body or upper body musculature. Fourth, despite our best efforts to verbally encourage all participants to train to momentary concentric failure, some volitionally stopped short of this directive during training. Participants in REPS appeared to have greater difficulty approaching true failure on average, likely due to greater metabolic acidosis and discomfort. That said, all subjects trained with a high level of effort throughout the study period, which has been shown to be sufficient for maximizing muscular adaptations (Grgic et al., 2021); thus, the degree of effort likely did not influence results between conditions. Future work may wish to obtain ratings of perceived effort and/or repetitions in reserve to directly evaluate subjective estimates of proximity to volitional failure. Fifth, although all subjects had previous RT experience (at least 1 year of consistent lower body RT), their experience varied across the cohort, and as a group, they would not be considered highly trained individuals. Thus, the sample would be more reflective of the average regular gym-goer and results therefore cannot necessarily be generalized to elite athletes and high-level bodybuilders. Moreover, previous squat experience was not a requirement of the study and many of the subjects did not regularly include squats in their training routines. Thus, some of the gains in dynamic strength conceivably can be attributed to initial neuromuscular improvements and may not reflect what would be achieved by those who squat on a regular basis. Finally, our findings are specific to a relatively short training block (8 weeks); it remains questionable as to whether and how results might be influenced by continuing the intervention over a longer timeframe. That said, many individuals plan their training programs in mesocycles lasting several weeks to months, making the results highly practical from a prescription standpoint.

Conclusion

Progressing load and repetitions throughout an 8-week training cycle produced similar increases in muscle size in most muscles and regions of the lower body. This suggests that both are likely sufficient for maximizing hypertrophy, at least in the short to medium term. However, we found modestly favorable aggregate MT measures favoring RF growth in REPS. Thus, it is possible that using repetition progressions is favorable in some contexts over others, but this requires replication and future work. Load progressions were slightly more effective for maximal strength and equally effective for muscular endurance performance. Further studies are needed to help decipher when, how, and for what populations different methods of progression should be employed to optimize muscular adaptations. However, from this work, it seems progressively increasing repetitions may be another option that trainees can use to improve their strength and muscle size, which is particularly useful when greater loads may not be available.

Supplemental Information

Supplemental Information 1 Raw Data.

Click here for additional data file.

Supplemental Information 2 Leave-one-out sensitivity analysis of all outcomes.

We re-ran all analyses after excluding each participant, one at a time. This assessed the influence of each participant on the estimated treatment effect. There were some instances where individual participants were indeed influential (e.g., MG muscle thickness), but none of these instances were enough to alter our conclusions.

Click here for additional data file.

We are grateful for the help of the following research assistants in conducting data collection: Carl Williams, Avery Rosa, Julia Torregrossa, Francesca Augustin, Hugo Zambrano, Xavier Torres, Astrid Jimenez, Roberto Arias, Mariella Mercado.

Additional Information and Declarations

Competing Interests

Author Contributions

Human Ethics

Data Availability

Brad J Schoenfeld serves on the scientific advisory board of Tonal Corporation, a manufacturer of exercise equipment. Mike Israetel and Jared Feather are employed by Renaissance Periodization. The other authors declare no competing interests.

Daniel Plotkin conceived and designed the experiments, performed the experiments, authored or reviewed drafts of the article, and approved the final draft.

Max Coleman performed the experiments, authored or reviewed drafts of the article, and approved the final draft.

Derrick Van Every performed the experiments, authored or reviewed drafts of the article, and approved the final draft.

Jaime Maldonado performed the experiments, authored or reviewed drafts of the article, and approved the final draft.

Douglas Oberlin performed the experiments, authored or reviewed drafts of the article, and approved the final draft.

Michael Israetel conceived and designed the experiments, authored or reviewed drafts of the article, and approved the final draft.

Jared Feather conceived and designed the experiments, authored or reviewed drafts of the article, and approved the final draft.

Andrew Alto performed the experiments, authored or reviewed drafts of the article, and approved the final draft.

Andrew D. Vigotsky conceived and designed the experiments, analyzed the data, prepared figures and/or tables, authored or reviewed drafts of the article, and approved the final draft.

Brad J. Schoenfeld conceived and designed the experiments, performed the experiments, prepared figures and/or tables, authored or reviewed drafts of the article, and approved the final draft.

The following information was supplied relating to ethical approvals (i.e., approving body and any reference numbers):

The Lehman College Institutional Review Board approved the study (IRB# 2021–2132).

The following information was supplied regarding data availability:

The raw data for outcomes are available as a Supplemental File.

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
