# Peer review of "Progressive overload without progressing load? The effects of load or repetition progression on muscular adaptations"

_PeerJ, doi:10.7717/peerj.14142_

## Round 0.1 · original submission · Major Revisions

Although the article has merit and potential, significant revisions should be performed considering the reviewers' comments.

·

Basic reporting

See below

Experimental design

See below

Validity of the findings

See below

Additional comments

Overall, this is an excellent study. The question is interesting, and the design, methods and analysis are great. I do have several suggestions which I think can improve the readability of the study and some of the discussion points.
Major points:
I think the manuscript would benefit from rewriting sections from the discussion section which currently focus on attempting to explain the negligible differences between conditions rather than discussing the similarities and the consequences of these similarities. That is, trainees can select an approach they prefer.
TITLE
Maybe skip the first part the ends with the question.
ABSTRACT
7 – What exercises did they perform?
14 – Seems rather uncertain, no? Maybe add a hedge. Also, you should clarify that there was a time effect in which both groups improved at all outcomes. It is not clear from the text.
INTRO
20- Consider removing the first sentence.
27- all current models? Maybe most?
31-33 – not very clear and I suggest some refs.
METHODS
98 – Was the upper body program the same for both groups or also followed the load and rep progression?
101 – why 10 and not 1RM? This is a little confusing to me. I suggest moving this part elsewhere.
105- what do you mean by maintaining? What were the precise instructions? Aim for 12 but at least 8? RIR based? Failure based?
177 –If this took place after anthropometrics then that means they did not eat anything for roughly 9 hours prior to these tests. If I am correct, please note this and the possible implications.
180- I suggest 1RM rather than repetition maximum testing. I got confused by this and thought this was the reps to failure test.
193 – if that is the case then remove it from the abstract.
207 – this is commendable but maybe also acknowledge that it was impossible to do in the strength test thus some bias may have been introduced.
213 – the citation (17) is concerned with comparing the p values of pre-post results of the two groups, but I am not sure why not include a single test across the two groups to provide the reader with an indication of the extent of improvement over time. While not the time effect is not the primary outcome, the primary outcome (between-group diff) can and should be interpreted in view of the magnitude of the time effect. That is, if both groups barely improved over the intervention then the similarities between them would tell a completely different story compared to, say, a large and similar improvement over the intervention.
255- why %? Stick to the number of reps I would think.
267- as per my point about the time effect, here you begin with a quick note about the small effects over time which provides context to interpret the between-group effects. I suggest doing this across all outcomes, even without a statistical test.
DISCUSSION
305- a third likely option that should be made explicit is that this finding was a random occurrence as it seems very unlikely that just the RF would respond in such a matter in contrast to all muscle groups tested.
310- again, the 2 kg diff between-group diff would be much easier to interpret and understand if provided in the context of the overall improvements across groups in the squat. Also, when inspecting the graph there seems to be a person who improved their 1Rm by 55+ kg. I presume that if you used a leave one out analysis then the favorable outcome would disappear or at least shirk considerably. Maybe include a leave-one-out analysis as I see a similar sort of outlier in the endurance graph.
315- the speculation can be a little confusing as you previously acknowledged that the effect is likely negligible. If anything, what requires explaining is the similarities in effects rather than why the load condition led to improved performance.
319 – the part about the 10rm was and still is unclear to me. Was it calculated somehow via the 1RMs?
415 – I think you should “celebrate” the similarities a bit more. Most of your discussion revolves around speculating about the negligible differences. Yet, the similarities – which were clear and consistent - have meaningful and practical consequences. Mainly, they allow choice. There doesn’t seem to be an optimal way so trainees can select a progression model is that more aligned with their preferences and perhaps access to equipment. Choices are good and people like them. This should be stated with clarity. In fact, I would place a much greater emphasis on that in the discussion than the small and arguably meaningless differences between conditions.

·

Basic reporting

I'm not the best person to mention something about English, and the other notes are attached.

Experimental design

There are some points to be adapted, as shown in the annex

Validity of the findings

There are some points to be adapted, as shown in the annex

Additional comments

There are some points to be adapted, as shown in the annex

Reviewer 3 ·

Basic reporting

First of all, I appreciate the opportunity to review this work investigating a relevant with information that will assist coaches during resistance training prescription. The text was written clearly and with appropriate language. The paper was correctly structured with figures prepared according to the journal's guidelines. Raw data were also provided by the authors.

Experimental design

The main objective of the present study to compare the effects of load increases while keeping repetition range constant versus increasing repetitions while keeping load constant on lower body muscle hypertrophy, muscle strength, and muscle endurance in resistance-trained individuals over an 8-week study period. Overall, the work was well written and minor revisions should be made to make it ready for publication Briefly, the reasoning presented in the introduction needs to be revised and restructured for a better theoretical foundation of the study problem. In addition, I suggest that the study hypothesis be included according to the objective presented. Some methodological procedures used for measurements of muscle thickness and maximum strength should be better described, so that they are replicable (see detailed comments). This will facilitate the understanding of the observed results. The item "statistical analysis" was presented in a clear and detailed way. The innovative statistical procedures used are in line with the purpose of the study.

Validity of the findings

The results were also well described, with the use of graphs and tables that allowed an adequate understanding of the responses verified for all analyzed variables. Finally, some parts of the discussion need to be rewritten to improve the reasoning used to justify the results verified in this study. The conclusion was presented concisely, making clear the practical applications of the verified results, as well as the limitations. Therefore, I suggest that the authors carefully read and respond to the specific comments presented below.

Additional comments

Main comments:

1 - Lines 55-57: As reported in the general comments, although important information was presented to introduce the investigated topic, a better theoretical basis would be needed for the study problem. In addition, I recommend that the study hypothesis be presented.

2 - Lines 87-94: The control group is known to be an essential part of the research (mainly using an experimental design between subjects) as it allows researchers to minimize the effect of all variables except the independent variable. I suggest that the authors present the reasons for not including a control group in this study. In addition, report on paper the procedures adopted to minimize the lack of control group.

3 - Lines 149-143): It has been reported that muscle thickness measurements acquired by ultrasound images are highly dependent on the procedures used to identify the measurement sites. Therefore, it is important that these procedures are standardized to allow replication in post-test measurements. In this sense, please provide a more detailed description of these procedures and equipment used to identify the sites on the thigh and leg, consequently to measure the muscle thickness of the participants. If possible, I suggest that ultrasound images be added to exemplify the analysis procedures performed, also that they be replicated in future studies.

4 - Lines 296-308: The reasoning presented to justify the modest superiority for increases in summed rectus femoris thickness for the repetition group is unclear. The arguments presented are based on increased residual fatigue of the vastus muscles at the end of a set during repetition progression. However, load progression was also performed until muscle failure and with moderate training volume (8-12 RM). Therefore, it is necessary that this information be reformulated so that it can better explain the rectus femoris muscle thickness outcomes.

Specific comments:

5 - Lines 57-58: Please cite references to support the information presented. “Current evidence…”.

6 - Line 111: The term cadence is used in the general sense of movement. Specifically, it can be understood as rhythm or regular succession of movements. In resistance training, it would be the succession of repetitions performed in the prescribed exercises, making it necessary to previously define the repetition numbers, which did not occur for the REP group. Also, the cadence concept does not determine the duration of each muscle action or repetition. Therefore, I suggest the use of the term concentric/eccentric duration or tempo, consequently repetition duration or repetition tempo.

7 - Lines 111-113: How were the durations of muscle action controlled throughout the training sessions? Did you use the metronome for this? If so, was it possible to maintain the predetermined concentric and eccentric durations with just the metronome when participants were close to muscle failure?

8 - Lines 135-140: Given that water status can influence body composition measurements, describe how the volunteers were prepared for the electrical bioimpedance analysis. Commonly, the participant cannot drink water for at least 4 hours before taking the body composition measurements. See below the studies used to support the comment.

https://pubmed.ncbi.nlm.nih.gov/8780358/

https://pubmed.ncbi.nlm.nih.gov/8039483/

9 - Lines 191-192: Do ICC and CV values refer to 1 RM performance in the squat exercise executed in different experimental sessions (test-retest condition)? If so, what would be the relevance of this information for the procedure used in present study, since only one experimental session was made available for the 1RM test? If more experimental sessions were used to perform 1RM tests on baseline measurements, would there be an expectation of an increase in dynamic muscle strength? If so, it would be a limitation of the procedure adopted. See below the studies used to support the comment.

https://www.scielo.br/j/rbme/a/w3KnY8pk9fQxy7qV5L39jfP/?format=pdf&lang=en

https://pubmed.ncbi.nlm.nih.gov/19855328/

10 - Lines 201-202: How was the range of motion (90 degrees of leg flexion to 0 degrees) controlled during the muscular endurance test?

11 - Lines 395-397: In future studies investigating this topic, it would be interesting to suggest the use of the repetitions in reserve scale. This procedure would allow researchers to more clearly identify when participants did not perform sets until muscle failure. Thus, they could have greater control of the effort demand during training sessions.

---

## Round 0.2 · Minor Revisions

Two reviewers provided annoted manuscript with minor changes to perform during this new round of revisions.

·

Basic reporting

See comments in word file.

Experimental design

See comments in word file.

Validity of the findings

See comments in word file.

Additional comments

See comments in word file.

·

Basic reporting

General comments:
Title
The title is creative, however, it makes it difficult to understand and especially to understand the study.
Abstract
Are presented satisfactorily.
Introduction
Are presented satisfactorily.

Experimental design

Methods
It should present more clearly the design of the study.
The design should come at the beginning of the methodology as a topic to illustrate what was actually done.
Results
Are presented satisfactorily.
Discussion
Are presented satisfactorily.
Conclusion
Are presented satisfactorily.
References
Are presented satisfactorily.
Overview
The manuscript presented addresses a relevant research topic.
It would be advisable to do a general review.

Validity of the findings

The presented version is ready to be published

Additional comments

The presented version is ready to be published

·

Basic reporting

I thank the authors for responding in detail to all comments. I am satisfied with the answers presented and I decide to accept the paper. Furthermore, I would like to congratulate all authors for the relevance and methodological quality of this work. I appreciate the opportunity to review this paper.

Experimental design

All comments submitted for this topic have been answered properly.

Validity of the findings

All comments submitted for this topic have been answered properly.

Additional comments

I have no additional comments.

---

## Round 0.3 · accepted · Accept

I can confirm that the authors significantly improved the manuscript, strictly following the reviewers' suggestions. The article presents merit and deserves a space in the publications of PeerJ.

·

Basic reporting

NA

Experimental design

NA

Validity of the findings

NA

Additional comments

Great work with the revisions.

·

Basic reporting

I'm not the best at reviewing English. References have been updated. The structure and results were presented satisfactorily.

Experimental design

The experimental design of the manuscript was presented satisfactorily.

Validity of the findings

The findings are presented clearly and concisely. I find it satisfactory.

Additional comments

In view of the adjustments that were made, I consider the manuscript in condition for publication in the form it is.